# Development of Ic-ELISA and Colloidal Gold Lateral Flow Immunoassay for the Determination of Cypermethrin in Agricultural Samples

**DOI:** 10.3390/bios12111058

**Published:** 2022-11-21

**Authors:** Lianrun Huang, Fuxuan Zhang, Fangxuan Li, Yin Jia, Minghua Wang, Xiude Hua, Limin Wang

**Affiliations:** 1College of Plant Protection, Nanjing Agricultural University, Nanjing 210095, China; 2State and Local Joint Engineering Research Center of Green Pesticide Invention and Application, Nanjing 210095, China

**Keywords:** cypermethrin, monoclonal antibody, colloidal gold, lateral flow immunoassay

## Abstract

Cypermethrin (CYP) is an insecticide in the pyrethroid family and is used widely in agriculture and for public health purposes. However, CYP has been shown to have negative impacts on reproduction, immunity and nerves in mammals. In this study, a monoclonal antibody (mAb) against CYP was prepared and used to establish an indirect competitive immunosorbent assay (ic-ELISA) and colloidal gold lateral flow immunoassay (LFIA) for the quantitative and qualitative determination of CYP residues in agricultural products. The half inhibition concentration of the ic-ELISA was 2.49 ng/mL, and the cut-off value and visual limit of detection of the LFIA were 0.6 and 0.3 μg/mL, respectively. The recovery rates of the ic-ELISA ranged from 78.8% to 87.6% in tomato, cabbage and romaine lettuce. The qualitative results of LFIA and quantitative results of ic-ELISA and HPLC were in good agreement in blind samples. Overall, the established ic-ELISA and LFIA proved to be accurate and rapid methods for the determination of CYP in agricultural products.

## 1. Introduction

Cypermethrin (CYP) is a common insecticide belonging to the pyrethroid family. It is widely used in agriculture and for public health purposes [1]. The toxic effects of CYP on reproduction, immunity and nerves in mammals are well-documented, and the World Health Organization has classified it as a moderately toxic compound [2,3,4,5]. The increased use of CYP has led to the widespread detection of residual CYP in soil, water and food [6,7,8]. These toxic substances may enter the body through the food chain and pose a threat to human health [9,10]. Given the widespread use and toxicity of CYP, several countries have established maximum residue limits (MRLs) for CYP. For example, the MRL range of CYP in the European Union is between 0.05 mg/kg and 2 mg/kg in products (Reg. (EU) 2017/626). In China, the MRLs of CYP in different agricultural products range from 0.01 to 7 mg/kg (National Food Safety Standard—Maximum residue limits for pesticides in food GB 2763-2021). It is important to determine the presence of CYP residues not only to ensure the safety of agricultural products, but also to protect human health.

Currently, CYP residues in various samples are mainly detected by instrument methods, including high-performance liquid chromatography (HPLC) and liquid chromatography tandem mass spectrometry (LC-MS/MS) [11,12]. These well-established chromatographic methods demonstrate excellent detection limits and reliable accuracy for effective monitoring of CYP residues. However, these methods require cost-prohibitive instrumentation, prolonged test time and trained staff, and are not suitable for rapid field screening.

The alternative to instrument methods is immunoassays, such as enzyme-linked immunosorbent assays (ELISAs) and lateral flow immunoassays (LFIAs) [13,14,15]. Based on antibody–antigen interactions, ELISA methods have been used for the detection of pyrethroid residue with the advantages of high sensitivity, rapidity, low cost, high-throughput and suitability for on-site analysis [16]. Xu et al. established indirect competitive ELISA (ic-ELISA) based on a monoclonal antibody for CYP determination with a half inhibition concentration (IC_50_) of 1.7 ng/mL [17]. Tu et al. prepared an antibody against CYP to develop an immunoassay with an IC_50_ of 2.6 ng/mL [18]. Meanwhile, LFIA has become increasingly popular for the determination of CYP due to its one-step operation and portable and naked-eye readout merits. Kranthi et al. developed a colloidal gold LFIA for the detection of CYP with a minimum detection concentration of 800 ng/mL [19]. Zhang et al. established an LFIA for the simultaneous detection of fenpropathrin, CYP, deltamethrin and cyhalothrin in vegetables and fruits. For CYP, the sensitivity of LEIA was 5000 ng/mL [20]. However, the sensitivities of the above LFIA methods cannot meet the actual requirements in practical applications.

Herein, CYP haptens were used to prepare anti-CYP mAbs. Then, ic-ELISA and colloidal gold LFIA based on the mAb were developed for the determination of CYP. The sensitivities and specificities of these immunoassays were evaluated. Recovery tests in three agricultural products were carried out to verify the accuracy of the developed immunoassays. Finally, the ic-ELISA and LFIA results were validated by HPLC in the analysis of the blind samples.

## 2. Materials and Methods

### 2.1. Reagents

Cypermethrin (98.5%), bifenthrin (98.3%), fenpropathrin (98.1%), ethothrin (98.4%), tefluthrin (98.3%), cyhalothrin (99.3%), cyfluthrin (98.5%), fenvalerate (98.3%) and deltamethrin (98.7%) were obtained from Shanghai Pesticide Research Institute Co., Ltd. (Shanghai, China). Haptens 1–7 (Table 1) were gifts from Professor Bruce D. Hammock [13,21]. ELISA plates were supplied by Jet Biofil (Guangzhou, China). For the assembling LFIA strip, nitrocellulose (NC) membrane, absorbent pad, sample pad and polyvinylchloride (PVC) plate were purchased from Shanghai Jiening Biological Technology Co., Ltd. (Shanghai, China). Chloroauric acid and goat anti-mouse IgG antibodies (GAMA) were obtained from Sigma-Aldrich (St. Louis, MO, USA). The optical density values of ELISA were detected by Spectra Max M5 from Molecular Devices (Sunnyvale, CA, USA). Aglient-1206 HPLC (Agilent Technologies, Inc., Santa Clara, CA, USA) was used to verify results of ELISA and LFIA.

### 2.2. Preparation of Antigens

The antigens were conjugated using diazotization and activated ester methods according to previous publications [16,22]. Because it can expose the unique portions of CYP and prevent haptens from folding onto the carrier protein, a short space arm of immunogen hapten proved to be helpful for mAb development [13]. Therefore, hapten 1 was conjugated to keyhole limpet hemocyanin (KLH) as immunogen. Haptens 1–7 were conjugated to bovine serum albumin (BSA) as coating antigen. The conjugates were identified by UV–Vis spectrophotometry, and their molar ratios of the hapten-to-carrier protein were estimated according to previous research [23].

### 2.3. Preparation of mAbs

The mice were immunized according to reported studies [24]. Five female Balb/c mice (6 weeks old) were injected subcutaneously with hapten 1-KLH emulsified with Freund’s adjuvant. After the fifth injection, the titer of the antiserum from immunized mice was detected by indirect ELISA. Mice with high titers were selected as spleen donors. The collected splenocytes and SP2/0 mouse myeloma cells were fused by PEG 1500 [25]. The positive hybridoma cells were screened by ELISA, and stable cell lines were established by the limiting dilution method [26].

To prepare monoclonal antibodies against CYP, the presensitive Balb/c mice were intraperitoneally injected with positive hybridoma cells (10 × 10^6^) to introduce ascites. The monoclonal antibodies were purified by Protein A columns according to the manufacturer’s instructions.

### 2.4. Ic-ELISA Procedure

The ic-ELISA procedure was executed according to previous studies [27]. Each well of the ELISA plate was coated with 100 μL of coating antigen diluted with carbonate buffer (0.05 mol/L, pH 9.6) at 4 °C overnight. After washing three times with phosphate-buffered saline (PBS, 0.01 mol/L, pH 7.4) containing 1‰ Tween-20, the wells were blocked with 200 μL of 5% skim milk powder in PBS at 37 °C for two hours. After the same washing procedure, 50 μL of CYP standard solution and 50 μL of mAb were pipetted into each well and incubated at 37 °C for one hour. After washing five times, 100 μL per well of 0.5‰ horseradish peroxidase-labeled goat anti-mouse IgG antibody in PBS was added and incubated at 37 °C for one hour. After the wells were washed five times, 100 μL per well of fresh substrate solution (0.1 mg/mL 3,3′,5,5′-tetramethylbenzidine and 0.024‰ H_2_O_2_ in 0.1 M citrate acid buffer) was added to react at 37 °C for 15 min. Then, 50 μL per well of 0.2 mol/L H_2_SO_4_ was used to stop the reaction, and the absorbances were read at 450 nm with a microplate reader.

### 2.5. Preparation of LFIA Strips

Colloidal gold particles were synthesized with trisodium citrate as a reducing agent according to our previous work [28]. For the preparation of colloidal gold mAb, 6.65 μg/mL anti-CYP mAbs were added to the colloidal gold solution adjusted to pH 8.2 using 0.2 mol/L K_2_CO_3_ with shaking at room temperature for 1 h. BSA (1%) was used to block free colloidal gold particles. After 1 h, the mixture was centrifuged at 12,000 revolutions per minute (RPM) for 15 min to remove free anti-CYP mAbs. The centrifuged deposits were resuspended using one-tenth of the original volume of sodium borate solution (0.01 mol/L) containing 2% BSA and 3% sucrose and preserved at 4 °C.

As depicted in Figure 1A, the LFIA strip was assembled into four components. The center section of a PVC plate was affixed to an NC membrane. The sample pad and absorption pad were stuck on both ends of the PVC plate, and both overlapped with the NC membrane by 1 mm. A total of 2 mg/mL of hapten 5-BSA and 0.5 mg/mL of GAMA were sprayed onto the NC membrane as the test line and control line, respectively, with 5 mm intervals by an automatic dispenser. After drying at 37 °C for 12 h, the fabricated cards were cut into 4 mm wide LFIA strips.

### 2.6. Principle of LFIA Strip

One hundred microliters of the CYP standard solution and 4 μL of the colloidal gold mAb solution were added into a micropore and mixed. After 5 min, an LFIA strip was inserted into the mixture solution. During the mixed solution migration from the sample pad to the absorbent pad, colloidal gold mAb was captured by the T and C lines, resulting in the appearance of color bands within 10 min. When the T line disappeared or the color intensity of the T line was lower than that of the C line, the result was positive (Figure 1B,C). When the color intensity of the T line was not less than that of the C line, the result was negative (Figure 1D).

### 2.7. Optimization of Ic-ELISA and LFIA

The working buffers of ic-ELISA and LFIA were optimized separately. Working buffers with different Na^+^ concentrations, pH values and methanol/acetonitrile/acetone/dimethyl sulfoxide contents were used to dilute CYP standards into a series of concentrations. Comparing the sensitivity of detection between different working buffers, the best parameters were determined.

### 2.8. Sensitivity of Ic-ELISA and LFIA 

Under the optimal conditions, a standard curve was constructed using a series of concentrations of standard solutions (50, 25, 12.5, 6.25, 3.13, 1.56, 0.78, 0.39 and 0.20 ng/mL) to assess the sensitivity of ic-ELISA. The IC_50_ and linear range (IC_10_–IC_90_) were calculated by a logistic equation established with Origin Pro 8.0.

The sensitivity of the LFIA strip was evaluated by a series of CYP standard solutions (0.7, 0.6, 0.5, 0.4, 0.3, 0.2, 0.1, 0 μg/mL). For qualitative analysis, the sensitivity of the LFIA strip was appraised by the visual LOD (vLOD) and the cut-off value [29]. The vLOD is the lowest concentration of CYP at which the lighter color of the T line is observed compared to the C line. The cut-off value is the minimum CYP concentration at which no color band appears on the T line.

### 2.9. Specificity of Ic-ELISA and LFIA 

To determine the specificity of the ic-ELISA, the cross-reactivities (CR) of some pyrethroid analogues (bifenthrin, fenpropathrin, ethothrin, tefluthrin, cyhalothrin, cyfluthrin, fenvalerate and deltamethrin) were evaluated. The CRs of the ic-ELISA were calculated with the following equation:CR (%) = (IC_50_ of CYP/IC_50_ of analogues) × 100%.

To test the specificity of the LFIA, series concentration standard solutions of analogues (bifenthrin, fenpropathrin, ethofenprox, tefluthrin, cyhalothrin, cyfluthrin, fenvalerate and deltamethrin) were prepared with the optimal buffer and analyzed by LFIA. The CRs of the LFIA were calculated with the following equation:CR (%) = (vLOD of CYP/vLOD of analogues) × 100%.

### 2.10. Analysis of Spiked Samples

Tomatoes, cabbage and romaine lettuce were all purchased from a local supermarket and identified as CYP-free by HPLC. Different pretreatment methods were used because the sensitivities and matrix interferences of different detection methods were different. The samples were minced and homogenized. We added 10 g of sample to a 50 mL centrifuge tube. For ic-ELISA detection, the samples were extracted with 20 mL of the optimal working buffer containing 60% methanol through vortex oscillation for 10 min and ultrasonic treatment for 5 min. After centrifugation at 4000 RPM for 5 min, all of the supernatant was collected and transferred to a 25 mL volumetric flask. The volume of extracted solution was accurately adjusted to 25 mL with the optimal working buffer containing 60% methanol for further dilution and detection.

For the LFIA analysis, 10 mL acetonitrile was added and vortexed for 10 min. After 10 min of ultrasound, 2 g NaCl and 3 g anhydrous Na_2_SO_4_ were added and fully oscillated for 10 min. After centrifugation at 4000 RPM for 5 min, 1 mL of the supernatant was taken and dried under nitrogen. The dried matrices were redissolved in the optimal buffer to be detected after proper dilution.

For the HPLC test, the extraction process was the same as the LFIA analysis, except the dried matrices were dissolved in 1 mL acetonitrile. The solution was passed through a 0.22 μm filter and detected by HPLC. HPLC detection conditions: XDB-C18 column (250 mm × 4.6 mm, 5 μm), mobile phase (acetonitrile: water = 90:10, *V*:*V*), flow rate of 1.0 mL/min, injection volume of 40 μL and detection wavelength of 225 nm.

## 3. Results

### 3.1. Identification of Antigens 

All hapten–carrier protein conjugates were identified with UV-Vis spectroscopy. UV-Vis spectra are shown in Appendix A. Compared with carrier proteins and haptens, the absorption peak of the conjugates at 280 nm was significantly shifted, indicating that the haptens were successfully conjugated to the carrier proteins. The coupling ratio for hapten 1-KLH was 12.1:1, and for hapten–BSA conjugates, it ranged from 1.3:1 to 17.1 (Appendix A).

### 3.2. Characterization of mAb

After five immunizations, the anti-serum of five mice was collected to determine the serum titer. All mouse serum had high titers ranging from 1:64,000 to 1:256,000. The mouse with the highest titer of 1:256,000 was sacrificed for cell fusion. After sub-clonal screening, four monoclonal cell lines capable of secreting anti-CYP antibodies were obtained, which were named 2G7H9, 2G9A8, 1E7D5 and 6F1G10. Then, the sensitivity of these monoclonal antibodies was evaluated by homologous ic-ELISA. The results are listed in Appendix A. The IC_50_ values of ELISAs based on these mAbs ranged from 19.14 to 575 ng/mL. The mAb 2G7H9 had the lowest IC_50_ of 19.14 ng/mL and was selected for subsequent studies. All hapten–BSA conjugates can be recognized by the anti-CYP mAb 2G7H9, and the IC_50_ values ranged from 5.75 to 19.14 ng/mL. Appendix A illustrates that the lowest IC_50_ (5.75 ng/mL) was obtained using hapten 5-BSA as the coating antigen, which was selected for subsequent research. The subtype of mAb 2G7H9 was determined as IgG2b by a kit. Then, 100 ng/mL and 50 ng/mL of hapten 5-BSA as coating antigen were used to calculate Kaff [30]. According to Appendix A, the Kaff value of mAb 2G7H9 was 3.64 × 10^8^ L/mol.

### 3.3. Identification of Colloidal Gold Labeled mAb

Gold nanoparticles with suitable size and good dispersion are helpful to improve the performance of the LFIA strip. The transmission electron micrograph of the colloidal gold solution (Appendix A) indicated that the synthesized gold nanoparticles were approximately 20 nm in diameter and had adequate dispersion. The result from Appendix A shows that the peak of colloidal gold mAb was shifted from 523 nm to 527 nm compared with colloidal gold, and the peak became wider. These results demonstrated the successful preparation of colloidal gold-labeled mAb.

### 3.4. Optimization of Ic-ELISA and LFIA

A_max_/IC_50_ (A_max_ was the absorbance of the negative control) was used to evaluate the performance of the ic-ELISA. After the checkerboard procedure (Appendix A), the optimum concentration of mAb and coating antigen was determined to be 0.5 μg/mL. Different organic solvents (10% methanol/acetonitrile/acetone/dimethyl sulfoxide) were used to improve the solubility of CYP in the working buffer. According to Appendix A, when methanol was the organic solvent, Amax/IC_50_ was the highest. With increasing methanol content (5%, 10%, 20%, 40%), the IC_50_ showed a trend of decreasing first and then increasing, and 10% methanol was selected as the optimal content. Similar experiments were performed to study the influences of Na^+^ concentrations (0.07, 0.14, 0.2, 0.3 and 0.4 mol/L) and pH (6, 7, 7.4, 8 and 9). When the concentration of sodium ions was 0.3 mol/L and the pH was 7.4, A_max_/IC_50_ was the highest. In summary, the optimal working buffer for ic-ELISA was 0.01 mol/L PBS (pH 7.4) containing 10% methanol and 0.3 mol/L NaCl.

In the optimization process of LFIA, the inhibition effect was evaluated by comparing the color intensities of the T lines when detecting CYP standard solutions with different concentrations (0, 0.3, 0.4 and 0.6 µg/mL). As shown in Appendix A, the best sensitivity of the LFIA was obtained when the Na^+^ concentration was 0.07 mol/L, and the optimal pH was determined to be 7. When the content of organic solvent was more than 20%, the color of the T line in the negative control was affected. When the buffer contained 10% dimethyl sulfoxide, the LFIA strip showed the best inhibition of CYP. Finally, the optimal working buffer solution was determined to be a phosphate buffer with pH 7 containing 0.07 mol/L NaCl and 10% dimethyl sulfoxide.

### 3.5. Sensitivity of Ic-ELISA and LFIA

Under the optimal conditions, the standard curve for CYP analyzed by ic-ELISA was established. As shown in Figure 2A, the IC_50_, linear range (IC_10_–IC_90_) and limit of detection (IC_10_) of ic-ELISA were 2.49 ng/mL, 0.40–7.87 ng/mL and 0.40 ng/mL, respectively. Compared with previous studies, the sensitivity of the established ic-ELISA was slightly lower than that of the ic-ELISA established by Xu et al. (IC_50_ was 1.70 ng/mL) and almost identical to that established by Tu et al. (IC_50_ was 2.59 ng/mL) [17,18]. 

As shown in Figure 2B, the vLOD and cut-off value of the LFIA strip were 0.3 and 0.6 μg/mL, respectively. In other words, if the detection concentration of CYP was lower than 0.3 μg/mL, the color of the T line was indistinguishably different from that of the C line, and the result was negative (-). If the CYP concentration was between 0.3 and 0.6 μg/mL, the color of the T line was visibly lighter than that of the C line, and the result was weakly positive (±). If the CYP concentration was ≥0.6 μg/mL, the T line color disappeared, and the result was positive (+). To our knowledge, the sensitivity of this colloid gold LFIA was better than that reported in the literature [19,20].

### 3.6. Specificity of Ic-ELISA and LFIA

Cyfluthrin, cyhalothrin, fenvalerate, deltamethrin, fenpropathrin, bifenthrin, ethofenprox and tefluthrin were used to evaluate the specificity (Figure 3A). As shown in Figure 3B, ic-ELISA showed cross reaction ratios of 31.09% and 16.15% for cyfluthrin and cyhalothrin, respectively. The specificity of the LFIA strip was similar to that of ELISA. LFIA strips showed negative results at a concentration of 20 μg/mL of structural analogues (CRs < 1.5%), except for cyfluthrin and cyhalothrin (Figure 3C). The visual detection limits of cyfluthrin and cyhalothrin in this LFIA were 1 μg/mL and 10 μg/mL and CRs were 30% and 3% (Appendix A). The reason for this may well exist in the delicate difference between their chemical structures. The structure of deltamethrin is similar to CYP, but the difference in halogen atom causes the inconspicuous cross-reaction. This result indicates that chlorine atoms play an important role in antibody antigen recognition, which is consistent with the results of Lee et al. [13].

### 3.7. Eliminated of Matrix Effects 

In practice, matrix effects influence the performance of immunoassays, leading to erroneous results. Usually, matrix effects can be eliminated by diluting the sample extract solution with the working buffer.

To eliminate the matrix effects on the ic-ELISA, the extracts of blank samples were diluted 15-fold, 30-fold and 60-fold with the optimum buffer to prepare a series of standard solutions for the construction of standard curves. When A_max_ and IC_50_ of the standard curve established using the diluted extracts were similar to those using the optimum buffer, the matrix effect of the sample was eliminated. The results in Figure 4A–C show that the matrix effects of tomatoes, cabbage and romaine lettuce were eliminated by dilution by 30-fold, 60-fold and 60-fold, respectively.

For the LFIA, extracts of blank tomatoes, cabbage and romaine lettuce were diluted 2-, 4- and 8-fold with the optimal buffer. These diluted extracts were spiked with a series of concentrations of CYP (0, 0.2, 0.3, 0.4, 0.5, 0.6 μg/mL) and tested using LFIA strips. The test results showed that the cut off value and vLOD of the LFIA in the 4-fold diluted tomato and cabbage matrix and the 8-fold diluted romaine lettuce matrix were the same as those in the optimal buffer (Figure 4D–E), indicating that the matrix effects were eliminated. 

### 3.8. Recovery

Recovery experiments were carried out to verify the viability of the immunoassays. For the ic-ELISA analysis, the agricultural products were fortified with 0.1, 0.2 and 0.4 μg/g. As shown in Table 2, the average recovery rates of the ic-ELISA ranged from 78.8% to 87.6%, with relative standard deviation (RSD) ranging from 0.3% to 3.1.

Tomato and cabbage samples were spiked with 0.75, 1.5 and 3 μg/g of CYP standards and romaine lettuce samples with 1.5, 3 and 6 μg/g for the recovery experiment of the LFIA. As shown in Table 2, when the spiked concentrations of tomato, cabbage and romaine lettuce were 0.75, 0.75 and 1.5 μg/g, respectively, all test results were negative. Exceeding the above concentration, the test results were positive. These results indicated that the FLIA strip had accurate qualitative detection results, and its limits of quantification on tomato, cabbage and romaine lettuce were 1.5, 1.5 and 3 μg/g, respectively.

According to GB 2763-2021, the MRLs for CYP in tomato, cabbage and romaine lettuce were 0.5, 5 and 7 μg/g, respectively. The established ic-ELISA and LFIA could meet the analytical requirements for CYP in these agricultural products except for using LFIA to detect CYP in tomato. This limitation could be circumvented by concentrating the extraction solution to increase the detection concentration.

### 3.9. Instrument Validation 

To further investigate the accuracy and reliability of the immunoassays, ten romaine lettuce samples were spiked to certain concentrations (this step was performed by a person who was not involved in testing). These samples were extracted as previously described and simultaneously detected by HPLC, ic-ELISA and LFIA. As shown in Table 3, the results of ic-ELISA and LFIA were consistent with those of HPLC. For quantitative analysis, the *t* test result (*p* > 0.05) showed that the results of ic-ELISA agreed with the detected data of HPLC fairly well. In qualitative detection, because of the 8-fold dilution to remove matrix effects, the LFIA analysis only generated positive results in samples 9 and 10, and no false positive results. These results indicated that the developed immunoassays were credible and feasible analytical methods for CYP.

## 4. Conclusions

In summary, mAb 2G7H9 was obtained by immunization of mice and cell fusion and used to develop ic-ELISA and LFIA. Under optimal conditions, the sensitivity of ic-ELISA was similar to the best sensitivities reported [17,18]. Meanwhile, the vLODs of the LFIA were superior to those of the reported colloidal gold strip methods for the determination of CYP [19,20]. Although less sensitive than ic-ELISA and HPLC, LFIA is an instrument-free analysis, and its results can be obtained within 15 min. These characteristics are more conducive to the field detection of a large number of samples. Furthermore, the results of the recovery and instrument validation experiments illustrated the accuracy and reliability of the established immunoassays for the determination of CYP in agricultural products. Overall, the ic-ELISA and LFIA developed in this study are sensitive, accurate and rapid methods for the monitoring of CYP in agricultural products. We provided an anti-CYP mAb with high sensitivity in this research. As the most core reagent in immunoassays, it can derive a series of immunoassays and collaborate with novel peptide ligands to further improve the analytical performances [31]. The proposed ic-ELISA and LFIA show high sensitivity and accuracy and are effective supplements to the toolbox for the detection of CYP in agricultural products.

## Figures and Tables

**Figure 1 biosensors-12-01058-f001:**
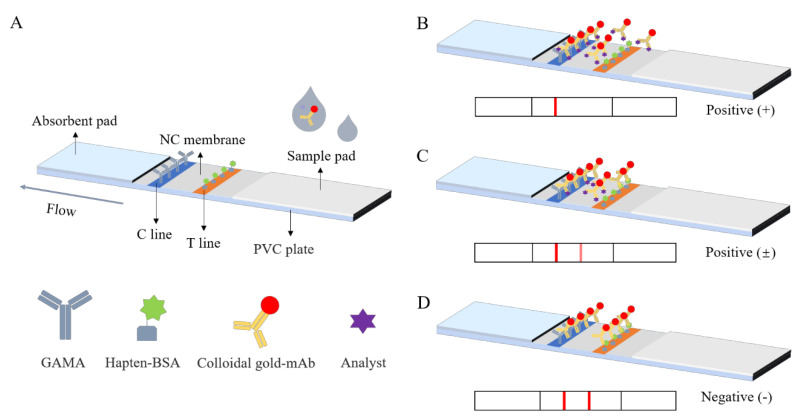
The illustration of the LFIA for CYP determination. (**A**) The structure schematic of the LFIA. (**B**–**D**) The different results of the LFIA.

**Figure 2 biosensors-12-01058-f002:**
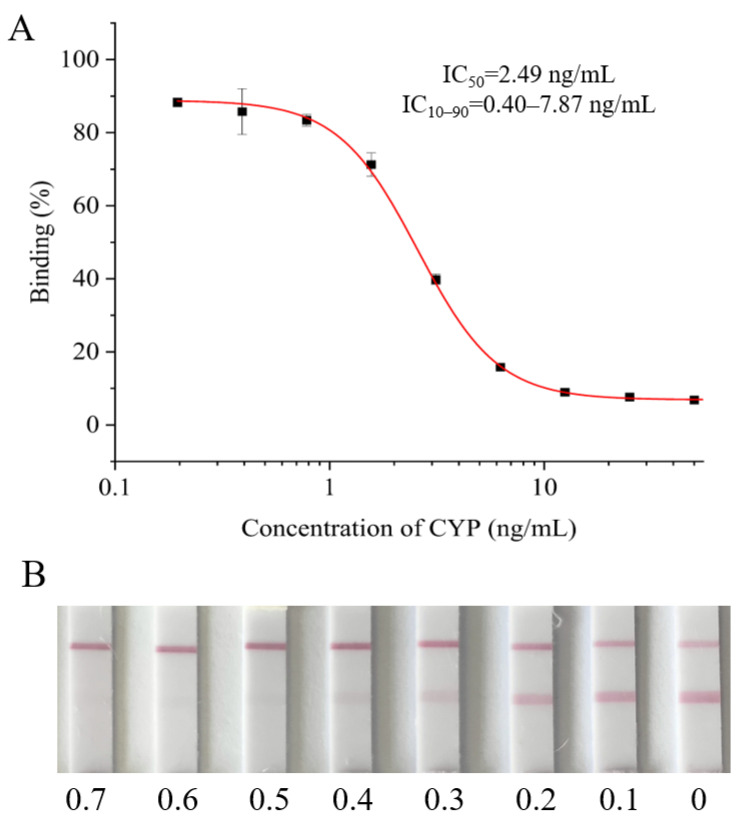
The sensitivity of ic-ELISA and LFIA. (**A**) The standard curve of ic-ELISA for CYP determination. (**B**) The sensitive analysis of LFIA; the concentrations from left to right are 0.7, 0.6, 0.5, 0.4, 0.3, 0.2, 0.1 and 0 μg/mL.

**Figure 3 biosensors-12-01058-f003:**
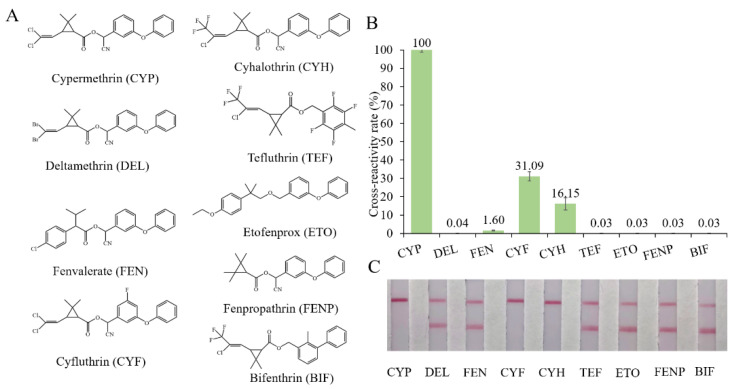
The cross-reactivity of the ic-ELISA and LFIA. (**A**) The cross-reactivity rates of ic-ELISA. (**B**) The specificity of LFIA. (**C**) The chemical structures of CYP and its analogues.

**Figure 4 biosensors-12-01058-f004:**
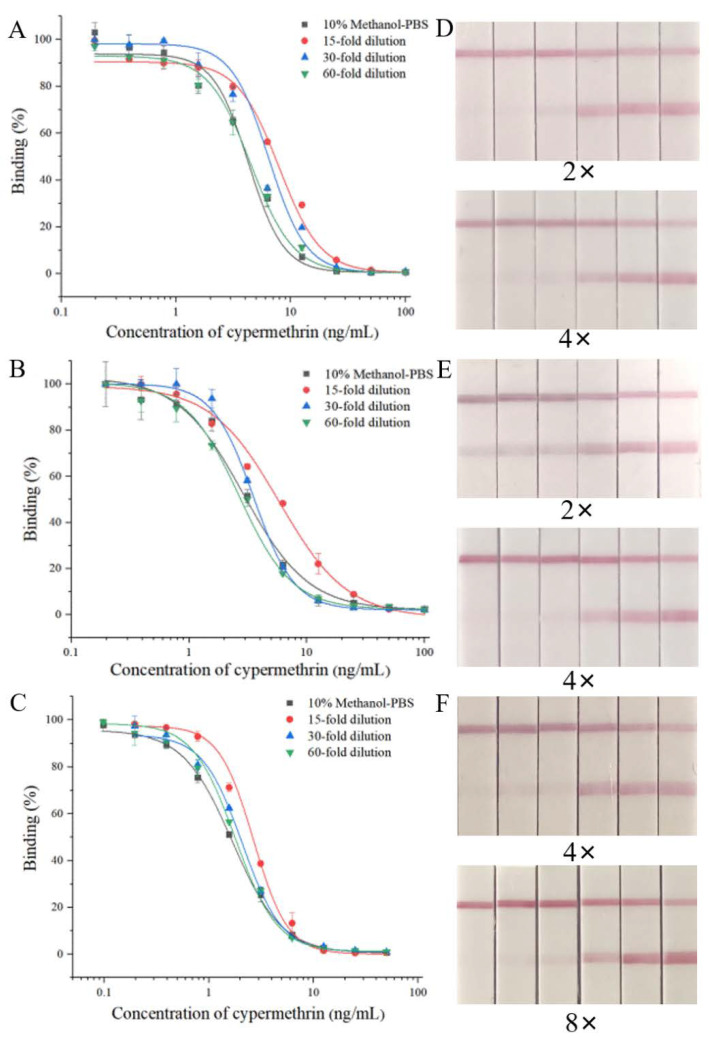
The matrix effects of ic-ELISA and LFIA. The matrix effects on ic-ELISA: (**A**) cabbage; (**B**) tomato; (**C**) romaine lettuce. The matrix effects on LFIA: (**D**) cabbage (**E**) tomato; (**F**) romaine lettuce; the concentrations from left to right are 0.6, 0.5, 0.4, 0.3, 0.2 and 0 μg/mL.

**Table 1 biosensors-12-01058-t001:** The chemical structures of haptens of cypermethrin.

Hapten	Chemical Structure	Reference
CYP	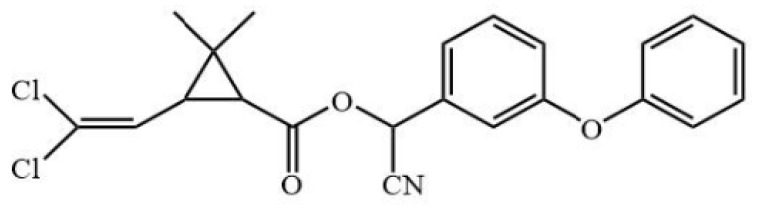	/
Hapten 1	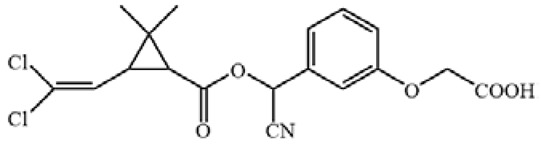	[13]
Hapten 2	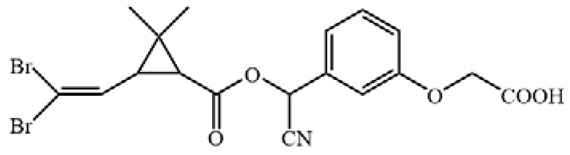	[13]
Hapten 3	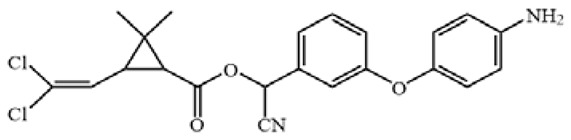	[13]
Hapten 4	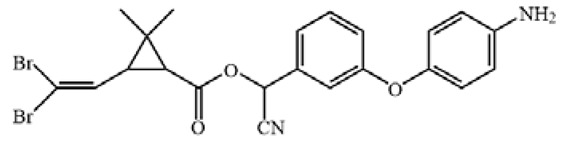	[13]
Hapten 5	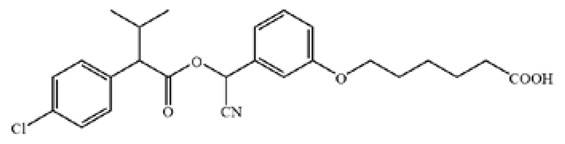	[21]
Hapten 6	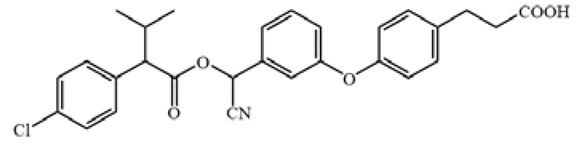	[21]
Hapten 7	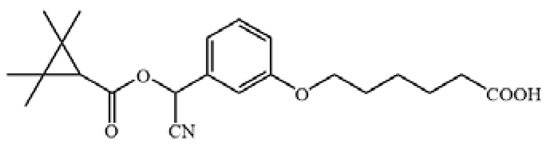	[21]

**Table 2 biosensors-12-01058-t002:** Recovery of CYP in the spiked samples by ic-ELISA and LFIA (*n* = 3).

Sample	ic-ELISA				LFIA			
	Spiked (μg/g)	Test Value ±SD (ng/g)	Recovery (%)	RSD (%)	Spiked (μg/g)	Result		
Tomato	0.1	81.4 ± 1.8	81.4	1.5	0.75	- ^a^	-	-
	0.2	161.8 ± 4.9	80.9	2.0	1.5	± ^b^	±	±
	0.4	330.4 ± 12.1	82.6	2.5	3	+ ^c^	+	+
Cabbage	0.1	78.8 ± 1.0	78.8	0.8	0.75	-	-	-
	0.2	160.6 ± 5.2	80.3	2.1	1.5	±	±	±
	0.4	349.2 ± 1.4	87.3	0.3	3	+	+	+
Romaine Lettuce	0.1	81.5 ± 3.8	81.5	3.1	1.5	-	-	-
	0.2	170.4 ± 3.8	85.2	1.6	3	±	±	±
	0.4	350.4 ± 13.2	87.6	2.9	6	+	+	+

^a^: “-” represents negative result; ^b^: “±” represents weakly positive result; ^c^: “+” represents positive result.

**Table 3 biosensors-12-01058-t003:** The real sample analysis by HPLC, ic-ELISA and LFIA (*n* = 3).

Method	Sample (ng/g)									
	1	2	3	4	5	6	7	8	9	10
HPLC	191.1	318.5	337.9	485.3	527.4	652.4	813.5	1353	3212	3511
ic-ELISA	204.8	300.1	352.6	450.4	545.3	615.2	821.3	1344	3222	3503
LFIA	- ^a^	-	-	-	-	-	-	-	± ^b^	±

^a^: “-” represents negative result. ^b^: “±” represents weakly positive result.

## Data Availability

The data presented in this study are available on request from the corresponding author. The data are not publicly available due to ethical constraints.

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
