# Peer review of "Development of Ic-ELISA and Colloidal Gold Lateral Flow Immunoassay for the Determination of Cypermethrin in Agricultural Samples"

_biosensors, 2022, doi:10.3390/bios12111058_

Round 1

Reviewer 1 Report

Dear Author, your manuscript was very well written and your findings clearly presented. 

Please see my comments below, 

Line 31 “ MRL range 30 of CYP in the European Union was between 0.05 mg/kg and 2 mg/mL in products” – the units are different so this statement is incorrect.

Line 43 E in Enzyme does not need to be in capitals.

Line 107 – problem with the symbol

What ethical approval was required to work with the mice? 

Author Response

(1)  Line 31 “MRL range 30 of CYP in the European Union was between 0.05 mg/kg and 2 mg/mL in products” – the units are different so this statement is incorrect.

Response: We really appreciate the reviewer’s comments. Changed as suggested. “2 mg/mL” has been changed to “2 mg/kg”.

(2) Line 43 E in Enzyme does not need to be in capitals.

Response: Changed as suggested. E in Enzyme in line 43 has been changed to lowercase.

(3) Line 107 – problem with the symbol

Response: Changed as suggested. The symbol “-” has been deleted.

(4) What ethical approval was required to work with the mice? 

Response: The procedures for care and use of the mice were approved by the Laboratory Animal Welfare and Ethics Committee of Nanjing Agricultural University and all applicable institutional and governmental regulations concerning the ethical use of animals were followed (SYXK(SU)2021-0086).

Author Response

  1. The cross-reaction formula in line 171 lacks a percent sign (%).

Response: We really appreciate the reviewer's comments. Changed as suggested. The percent sign (%) has been added in the cross-reaction formula.

  1. Line 172 to 173, the specific experiment description of LFIA is not clear enough, so it is suggested to add some specific explanations.

Response: Changed as suggested. The sentence has been added as “To test the specificity of the LFIA, series concentration standard solutions of analogues (bifenthrin, fenpropathrin, ethofenprox, tefluthrin, cyhalothrin, cyfluthrin, fenvalerate and deltame-thrin) were prepared with the optimal buffer and analyzed by LFIA. The CRs of the LFIA were calculated using the following equation: CR (%) = (vLOD of CYP / vLOD of analogues) × 100%” in 2.9 Specificity of ic-ELISA and LFIA. Meanwhile the sentence at line 285-289 has been modified as “The specificity of the LFIA strip was similar to that of ELISA. LFIA strips showed negative results to 20 μg/mL of structural analogues (CRs <1.5%), except for cyfluthrin and cyhalothrin (Figure 3C). Then the visual detection limits of cyfluthrin and cyhalothrin were measured as 1 μg/mL and 10 μg/mL, respectively, and the CRs were calculated as 30% and 3% (Figure S5)”.

  1. In line 259, “the vLOD and cut-off value of the LFIA strip were 0.6 and 0.3 μg/mL, respectively.”, I wonder if the values of these two parameters are reversed, please check and amend it.

Response: Changed as suggested. The sentence has been corrected to “the vLOD and cut-off value of the LFIA strip were 0.3 and 0.6 μg/mL, respectively”.

  1. In table 2, “Recovery±SD” should be replaced with “Test value±SD”, Please regulate the expression.

Response: Changed as suggested. “Recovery±SD” has been replaced with “Test value±SD” in table 2.

  1. In “Instrument validation”, Why only romaine lettuce samples were made? the instrumental validation data for tomatoes and cabbage should be added.

Response: Please allow us to make the following explanation: “The recoveries of spiked samples in ic-ELISA were range from 78.8% to 87.6% with RSDs≤3.1%, which can meet requirements (mean recoveries of 70 − 120% with CV ≤ 20%) of International Union of Pure and Applied Chemistry. Also, the recoveries in LFIA were consistent with the spiked concentrations. The results indicate that the proposed ic-ELISA and LFIA have satisfactory accuracies for CYP detection.

For further verifying the detection result, we carried out the instrument validation. Previous experiment showed romaine lettuce sample had stronger matrix effect than potato and cabbage (Figure 4), which means the tests on romaine lettuce samples are more likely to produce false results. So, the romaine lettuce samples were used as representative to further determine the accuracies of proposed immunoassays.

  1. The test result of LFIA in “Instrument validation” is inconsistent with the sensitivity of LFIA described above. Taking the result of sample 8 as an example, why the LFIA interpretation result was still negative when the drug concentration was 1353 ng/g?

Response: Changed as suggested. We have added the explanation at line 357-359 as: “In qualitative detection, because of the 8-fold dilution to remove matrix effects, the LFIA analysis only generated positive results in sample 9 and 10, and no false positive results.”.

  1. In the section of “Optimization of ic-ELISA and LFIA”, all of the optimizations were buffer optimizations under working conditions, are there any other optimization steps? could be appropriately supplemented.

Response: In addition to buffer optimizations, checkerboard titration was also used to optimize the concentrations of the coated antigen and antibody, as detailed in Supplementary material part 1. The optimum concentration of mAb and coating antigen was determined to be 0.5 μg/mL.

  1. In order to make the picture clear, it is recommended to remove the sample pad and absorbent pad from all the pictures of LFIA strips, and keep the C and T lines.

Response: Changed as suggested. The pictures of LFIA strips have been modified as required.

  1. The discussion and summary sections did not show the novelty of the experiment. It is suggested to add some discussion to strengthen the advantage.

Response: Changed as suggested. The discussion has been added to the end of conclusion part as: “We provided an anti-CYP mAb with high sensitivity in this research. As the most core reagent in immunoassays, it can derive a series of immunoassays and collaborate with novel peptide ligands to further improve the analytical performances. Besides, the pro-posed ic-ELISA and LFIA show high sensitivity and accuracy that are effective supplements to the tool box for the detection of CYP in agricultural products.”.